# Sleep Assessment in Patients with Inner Ear Functional Disorders: A Prospective Cohort Study Investigating Sleep Quality Through Polygraphy Recordings

**DOI:** 10.3390/audiolres15040076

**Published:** 2025-06-24

**Authors:** Dorota Kuryga, Artur Niedzielski

**Affiliations:** 1Clinic of Pediatric Otorhinolaryngology, Centre of Postgraduate Medical Education, 05-092 Dziekanow Lesny, Poland; artur.niedzielski@cmkp.edu.pl; 2Department of Otolaryngology, Bielanski Hospital, 01-809 Warsaw, Poland; 3Department of Otolaryngology, Voivodeship Hospital in Lomza, 18-404 Lomza, Poland; 4Department of Pediatric ENT, The Hospital’s Pediatric in Dziekanow Lesny, 05-092 Dziekanow Lesny, Poland

**Keywords:** labyrinth, perception, breathing, apnea, wakefulness

## Abstract

**Background/Objectives:** The vestibulo-respiratory reflex regulates the tension of the respiratory muscles, which prevents apneas and awakenings during sleep. This study aimed to determine whether functional deficits in the inner ear disturb sleep quality. **Methods**: We compared sleep parameters in patients with their first episode of acute inner ear deficit (Group A: sudden idiopathic vertigo attack, sudden sensorineural hearing loss), chronic functional inner ear impairment (Group B: chronic peripheral vertigo, permanent hearing loss), and in healthy individuals (Group C). Polygraphy recordings were performed twice, in Group A at the onset of acute otoneurological symptoms and the second time after their withdrawal with an interval of 1 to 13 days, in Group B after 1 to 6 days, and in Group C after 1 to 8 days. **Results**: In Group A during the symptomatic night, overall and central apnea-hypopnea indices were significantly higher and snoring time was longer. Group A also had higher central apnea-hypopnea index on the first night compared to healthy individuals. In chronic disorders, sleep recordings showed lower autonomic arousal index than in controls or symptomatic nights in Group A. **Conclusions**: These findings highlight the severity of sleep apnea indicators in Group A. Our results suggest that acute dysfunction of the inner ear substantially impacts central neuronal signaling responsible for regulating normal sleep-related breathing and leads to a deterioration in sleep quality in contrast to individuals with chronic inner ear impairments. It can also be assumed that people with chronic vertigo or hearing loss experience less interrupted sleep than healthy individuals.

## 1. Introduction

Adequate length and quality of sleep are essential for effective body regeneration, and proper breathing patterns greatly influence them. Apneic episodes and related hypoxemia lead to sleepiness, hypersomnolence, exhaustion, neurocognitive abnormalities, depression, and other terrifying consequences. The risk factors for obstructive sleep apnea (OSA) include age, male sex, obesity, and upper airway obstruction, whereas central sleep apnea (CSA) may be suspected in non-obese individuals without anatomical factors of upper airway obstruction [1]. CSA is characterized by a lack of drive to breathe during sleep and even if in the majority of cases chronic cardiological and neurological conditions are considered to be its cause, acute diseases (atrial fibrillation, systolic heart failure, stroke, brain trauma) can also lead to CSA events [2,3,4,5,6]. In polysomnography, concomitant CSA and OSA events are characteristic of various central nervous system disorders, e.g., Alzheimer’s disease, Parkinson’s disease, multisystem atrophy, amyotrophic lateral sclerosis. They are related to degeneration of the respiratory center nuclei, loss of motor neurons, chest wall rigidity, muscles hypokinesia, but also an imbalance of neurotransmitters, including orexin (hypocretin), which play a key role in regulating sleep and promoting wakefulness [7].

The vestibular system balances the activity of the autonomous nervous system, participates in breathing adjustments, and influences venous return during movements that could lead to unexpected orthostatic hypotension [8,9]. Vestibular nuclei (VNN) control ventilation parameters to ensure the delivery of adequate oxygen during movements and changes in posture and body position [10]. After stimulation of nerve VIII, the muscles of the tongue, larynx, and pharynx are involved in the vestibulo-respiratory reflex [11,12,13]. This is supposed to maintain airway patency and prevent the collapse of the tongue base during movement and gravity, especially when changing to the supine position, which is crucial during sleep to counteract airway obstruction in humans.

After inner ear (labyrinth) damage, neurophysiological consequences originating from the dysregulation of vestibulo-autonomic reflexes occur but stimuli from other organs can determine the compensation process, and some VNN neurons resume electrical activity, which allows the regulation of breathing and other autonomic reflexes [10,14]. Neuroplastic conditioning adaptation and compensation mechanisms occur during sleep, mainly rapid eye movements and slow waves [15].

Functional disorders of the inner ear frequently manifest after a night’s sleep, resulting in hearing loss when auditory cells are affected or vertigo when damage occurs to the semicircular canals and vestibulum. Sudden sensorineural hearing loss (SSNHL) is defined as a hearing loss of 30 dB or greater over at least three contiguous audiometric frequencies occurring within 72 h [16]. A sudden vertigo attack (SVA) can be described as an illusion of the world spinning around and is often accompanied by nausea, emesis, and other vegetative symptoms. Horizontal nystagmus is a classic objective sign of peripheral vertigo and is assessed using a simple visual evaluation or videonystagmography. Although inflammatory, vascular, or autoimmune backgrounds are widely considered, the etiology of these inner ear disorders remains unclear, with up to 90% of cases being idiopathic SSNHL [16].

Both hearing loss and vertigo appreciably worsen the quality of life, especially when they occur in young people or become irreversible, give the impression of disability, fear of falling, especially in older people, and are associated with stress, anxiety, and depression [17,18]. Some patients suffering from vertigo or hearing loss complain of various sleep-breathing disorders. In Taiwan OSA was reported in 1.7% of the population with SSNHL compared to 1.2% of the general OSA rate in patients without SSNHL [19].

This study aimed to assess the effects of a receptive functional deficit in the inner ear on the quality of sleep. We hypothesized that sensory loss of the inner ear may trigger changes in central neuronal signaling and consequently affect sleep-related breathing and thus sleep itself.

## 2. Materials and Methods

### 2.1. Setting

This study was set up before the COVID-19 pandemic; the research took place from 1 September 2018 to 31 October 2019 in the Department of Otolaryngology, Bielanski Hospital, Warsaw, Poland. The study was conducted in accordance with the Declaration of Helsinki and approved by the Ethics Committee of the Centre of Postgraduate Medical Education in Poland (approval No. 22/PB/2019; issued 13 March 2019).

### 2.2. Participants

A total of 60 participants were examined twice within 14 months. We prospectively compared three cohorts of patients, each comprising 10 males and 10 females. (1) Group A: idiopathic sudden functional inner ear disease for the first time in their life (SVA, SSNHL). Some patients reported vertigo combined with SSNHL, others vertigo with tinnitus or milder hearing impairment not meeting the criteria for SSNHL; (2) Group B: chronic functional inner ear disease (permanent hearing loss, chronic episodic peripheral vertigo, sometimes combined with chronic imbalance); (3) Group C: healthy population (no otoneurological complaints).

Recruited patients were selected based on the criteria of approximate age, sex, and body mass index (BMI). We focused on a population with a low risk of OSA or CSA to avoid over-adjustment for potentially undiagnosed disorders. Before the polygraphy, the patients were asked to avoid alcohol or psychoactive drugs. Exclusion criteria are presented in Table 1.

### 2.3. Sleep Recording Procedure

The patients included in the study underwent sleep registration using a SomnoCheck micro cardio device (Weinmann Medical Technology, Hamburg, Germany), a home sleep apnea testing (HSAT) portable type III monitor [1]. It used two sensors for oximetry and cardiac parameter measurement: a nasal cannula and a cardiovascular sensor. The measurements were limited to pulse wave amplitude and frequency, airflow, blood oxygen level, respiratory effort with chest and abdominal movements, and snoring (Sn).

The analysis of sleep time utilized a metric known as time with good flow signal quality (TGF). Apart from apnea hypopnea indicators (apnea-hypopnea index [AHI], central apnea-hypopnea index [cAHI], obstructive apnea-hypopnea index [oAHI]), the following were also used to assess sleep quality: autonomic arousal index (AAI), AAI caused by respiratory events like apnea or hypopnea (AAIre), AAI caused by increased respiratory effort (RERA), risk of sleep fragmentation, and Sn (Table 2). The minimum duration of one apnea or hypopnea episode was 10 s and of an arousal 3 s.

To determine the direct influence of acute disease on sleep in Group A, we compared the moment of symptom resolution with a relatively asymptomatic period. Hence, a comprehensive sleep test was conducted twice for each participant as part of the study protocol. The first assessment occurred on the night of admission to the ear, nose, and throat (ENT) department when symptoms were alleviated. The time criterion for HSAT recording in Group A was established for a maximum of 72 h from the onset of symptoms. The second evaluation occurred after the participant had reached a stable otoneurological state during an asymptomatic period, typically verified through an ENT examination within 1 to 13 days. By conducting two separate tests at different stages, we aimed to capture and compare the sleep patterns and quality during symptomatic and asymptomatic periods, thereby gaining a more comprehensive understanding of the effects of the participant’s condition on sleep. The ENT evaluations included otoscopy, tone and impedance audiometry, and visual nystagmus grading. Even if Groups B and C were expected to have stable inner ear conditions, a second test was also performed, after 1 to 6 and 1 to 8 days, respectively.

### 2.4. Statistical Analysis

We used the Wilcoxon signed-rank test and the *t*-test for independent samples for statistical purposes. The analysis was performed using the Statistica 13.1 software. A *p*-value of <0.05 was considered statistically significant.

## 3. Results

### 3.1. Characteristics of the Studied Groups

All examined groups were homogenous in number, sex, age, BMI, achieved records of Epworth sleepiness score, and in the interval between two nights of HSAT (Table 3).

### 3.2. Findings of Measured Parameters

#### 3.2.1. Sleep Time

Sleep shorter than six hours was recorded in 30% of individuals from Group A, in 15–20% from Group B, and in 5–15% of healthy people, depending on the tested night. The longest average TGF was registered in healthy controls, 6 h 58 min and 7 h 11 min depending on the night. The shortest sleep time was observed at the time of acute labyrinth shutdown (6 h 28 min), and it was, on average, 13 min shorter than after returning the function of the peripheral hearing and balance organ, 28–30 min shorter than in chronically ill patients, and 30–43 min shorter than in healthy people. These findings suggest poorer sleep during acute inner ear dysfunction. However, the time with good flow signal quality did not vary significantly between the groups or within the groups between analyzed nights (Table 4 and Appendix A).

#### 3.2.2. Apnea-Hypopnea Indicators

The highest average indicators of AHI and cAHI were recorded within Group A during active symptoms: AHI1A = 9.9 ± 10.3 and cAHI1A = 4.1 ± 3.2. Among the parameters that defined OSA/CSA, substantial changes were observed also only in Group A as illustrated in Figure 1a,b.

A statistically significant increase in apnea and hypopnea, and even more in its central component, was observed in patients presenting with acute inner ear dysfunction symptoms compared to those without (AHI1A = 9.9 ± 10.3 vs. AHI2A = 6.8 ± 4.4; *p* = 0.027) (cAHI1A = 4.1 ± 3.2 vs. cAHI2A = 3 ± 2.3; *p* = 0.011). Moreover, a higher cAHI was observed in the symptomatic period in Group A than in healthy subjects, both for the first registered night in Group C (cAHI1A = 4.1 ± 3.2 vs. cAHI1C = 2.3 ± 2.1; *p* = 0.049) and their second recordings (cAHI1A = 4.1 ± 3.2 vs. cAHI2C = 2.4 ± 1.4; *p* = 0.041). After symptom remission in Group A, the overall AHI and cAHI were comparable to controls and patients with chronic inner ear impairment. The obstructive components of apnea and hypopnea also decreased in Group A during the second night, but not significantly (oAHI1A = 5 ± 7.9 vs. oAHI2A = 3.1 ± 2.5). Although patients with chronic inner ear impairment had elevated AHI, cAHI, and oAHI values compared to controls, the differences between these groups and within them were not statistically significant, which is presented in Table 5, Figure 1a–c, and Appendix A.

#### 3.2.3. Sleep Fragmentation and Autonomous Arousals Indicators

The risk of sleep fragmentation was moderate in healthy individuals and in people with acute inner ear dysfunction on both measured nights and low in those with chronic inner ear impairment, with important differences between Groups B and C (*p* = 0.028; Appendix A).

Patients in Group A experienced more arousals during acute inner ear dysfunction than after recovery, and the values of AAI, AAIre, and RERA are shown in Table 6.

The highest average AAI was reported in Group A during the symptomatic period (AAI1A = 29.9 ± 17.9); it was slightly more frequent than in the healthy population (AAI1C = 27.3 ± 11.3, AAI2C = 28.5 ± 10.5) and the index was the lowest in Group B (AAI1B = 21 ± 11.3, AAI2B = 18.8 ± 9.9). Although AAIre and RERA were statistically comparable among all groups, individuals with chronic vertigo or hearing loss displayed the fewest of these types of arousals and awakenings. AAIre was most often observed in Group A on the night of admission to the hospital, on average seven per hour, whereas after symptom remission, it reached a value of 4.8 per hour, which was comparable to those of patients with chronic inner ear impairment (AAIre1B = 3.5, AAIre2B = 4) and the healthy population (AAIre1C = 4.5, AAIre2C = 4.9). Arousals related to respiratory effort were most often seen in controls (RERA2C = 2.1 ± 2.9), with the value comparable to a symptomatic night in Group A (RERA1C = 1.7 ± 2.9, RERA1A = 1.7 ± 2.2). RERA dropped in Group A after symptom remission to an average of 1.3 per hour, which corresponded to records of chronic inner ear deficit (RERA1B = 1.1 ± 1.9 and RERA2B = 1 ± 0.9).

AAI, AAIre, and RERA did not change within groups significantly over the two consecutive nights but differences were already noticeable between groups (Figure 1d–f). The number of spontaneous awakenings per hour of sleep was lower in patients with chronic symptoms than in healthy individuals (AAI1B = 21 ± 11.3 vs. AAI2C = 28.5 ± 10.5; *p* = 0.038), (AAI2B = 18.8 ± 9.9 vs. AAI1C = 27.3 ± 11.3; *p* = 0.015), (AAI2B = 18.8 ± 9.9 vs. AAI2C = 28.5 ± 10.5; *p* = 0.005), and those with acute symptoms (AAI2B = 18.8 ± 9.9 vs. AAI1A = 29.9 ± 17.9; *p* = 0.022) (Appendix A).

#### 3.2.4. Snoring Intensity

The most intense snoring was registered in healthy individuals and ranged from 6.3% to 7.4%. The least bothersome snoring was observed in Group A during the second night (Sn2A = 3.2%) and this subpopulation presented significantly less intense Sn after symptom regression (Sn1A = 5% ± 8.4 vs. Sn2A = 3.2% ± 6.4; *p* = 0.028). Such differences between the compared nights were not seen either inside Group B or inside Group C nor between particular groups (Table 7 and Appendix A).

## 4. Discussion

The vestibular organ is a strong non-photon circadian rhythm synchronizer. The absence of otolithic excitations in orexinergic neurons may disturb sleep–wake cycle transitions and reduce sleep time [15]; 45% of astronauts need sleeping pills, and sleep time during space missions remains reduced to approximately six hours [21,22,23]. In a US study the prevalence of sleep duration shorter than six hours was 15.5% in the population suffering from vestibular vertigo compared to 7.9% in the healthy population [24]. That matches with our findings, in which shorter sleep was found in 30% of individuals with acute symptoms of inner ear deficit and in 15–20% of chronically ill people compared to 5–15% of healthy individuals.

Many studies emphasize the connection between the vestibular system and respiratory control during head movements through the vestibulo-respiratory reflex. Fast-acting compensatory mechanisms allow the return of autonomic regulation of the respiratory systems during body movements after labyrinth damage through the connections between the VNN and the receptors of the somatosensory system; this compensation occurs effectively one week after labyrinthectomy in animals [14,25,26]. Human studies have confirmed that acute labyrinth dysfunction impairs the respiratory system’s response to head and body reorientation or rotational motion and is only associated with short-term effects on breathing [27]. In healthy individuals, stimulation of the vertical semicircular canal increases the respiratory rate by shortening breaths, suggesting its role in driving respiration under normal conditions [28]. A similar effect was obtained by stimulating the labyrinth with caloric tests in healthy individuals, whereas changes in breathing were not observed in individuals with persistent labyrinth dysfunction [29]. Our research was concentrated on nocturnal breathing and showed its disruption associated with acute hearing and balance receptor disease compared to an intact labyrinth. Our findings indicate that sudden inner ear dysfunction, but not chronic dysfunction, resulted in a higher occurrence of central and overall sleep apnea and hypopnea, and more intense snoring; this highlights the impaired regulation of breathing due to a weakened vestibulo-respiratory, reflex specifically during the initial night compared to the asymptomatic period. After otoneurological state stabilization, these parameters decreased to a level comparable to that in the control group and chronically ill patients. We suppose that compensatory processes in people after sudden inner ear loss influence the vestibulo-respiratory reflex and explain the statistically significant improvement in sleep polygraph parameters after symptom withdrawal, and the accomplishment of central compensation after approximately four days.

When differentiating the study groups, we focused on the duration of otoneurological symptoms (≤72 h vs. months/years) and on the presence (Group A) or absence (Group B) of vestibular nystagmus during the first HSAT. Group B patients were clinically underdiagnosed and discharged from the hospital with unilateral/bilateral sensorineural hearing loss and or “other peripheral vertigo”. They could then potentially suffer from benign paroxysmal positional vertigo, Ménière’s disease, as well as mild transient ischemic attacks or even psychogenic vertigo.

In the group of patients with a chronic inner ear impairment who had long undergone the compensation process, a comparable frequency of apneic episodes was observed between both nights, just like in healthy controls. However, we have detected a significantly lower risk of sleep fragmentation in chronically ill individuals compared to healthy people. They recorded significantly less general arousal than controls, and a similar trend was observed in AAIre and RERA. In addition, comparing the AAI between patients in Groups B and A, significantly fewer awakenings were found in chronically ill patients. These results diverge from other authors. Sowerby proved that patients with chronic idiopathic dizziness were likelier to have significant daytime somnolence than controls [30]. Martin et al. reported that patients with a history of chronic idiopathic bilateral vestibular loss (BVL) present higher nocturnal activity and worse sleep efficiency compared to that of healthy individuals [31]. He suggested that the amplitude of the activity–rest cycle is damped in BVL, which in the general population may indicates sleep fragmentation [32]. Chronic inner ear impairments cause all-day cognitive fatigue due to increased concentration requirements for sound perception and spatial orientation, which could potentially be associated with lower orexin levels and lower arousal indices.

A population-based study in the US examined the correlation between sleep apnea diagnosed with HSAT, defined by the cutoff AHI ≥ 15/h, and chronic hearing impairment described notably in the low-frequency range [33]. The authors did not confirm the variability in the intensity of snoring in connection with chronic hearing loss, which corresponds to our results; in the group of chronically ill patients, snoring remained at a stable level of 4–5% compared to the 6–7% in healthy individuals, with no significant difference between the two groups. However, the lack of labyrinth stimulation was associated with more intense snoring in Group A during the period of uncompensated disease compared to the second analyzed night.

Any acute stress, e.g., cranial trauma, small injury, anxiety attack, leads to the release of catecholamines and activates the hypothalamic–pituitary–adrenal axis. Norepinephrine, epinephrine, and cortisol increase the state of wakefulness and worsen sleep parameters. Signals from the limbic system, triggered by stress and emotions, affect the respiratory, sleep, wakefulness, hearing, and balance centers [34]. Anxiety can affect breathing, leading to hyperventilation in both healthy individuals and those with vestibular deficits; it can intensify the feeling of dizziness and instability, and in those with vestibular deficits additionally may cause nystagmus [35]. Due to internal masking factors such as stress and fear of disability, the results of our polygraphy recordings may be nonspecific for inner ear deficit. However, it was already reported that damage to the inner ear escalates the production of histamine, noradrenaline, and steroids, which can enhance the state of wakefulness and negatively affect the quality of sleep [15], especially since studies on rats have confirmed the bidirectional flow of information between vestibular nuclei and orexin neurons of the hypothalamus [36]. In addition, the polysomnography itself can also cause discomfort and stress, contributing to worse sleep parameters, which is described as the “first night effect” [37].

The major limitation of this study is the type of sleep study performed. The portable type III monitors used, with a limited number of recorded parameters, were insufficient to estimate and determine the physiological sleep staging. Nevertheless, the diagnostic accuracy of HSAT type III against full polysomnography in a low-risk OSA population like this examined in our work ranges from 70–78%, using a cutoff of AHI ≥ 5/h, which allows reliable data interpretation [1]. Furthermore, the absence of continuous monitoring by a trained supervisor prevented us from confirming if the time established for analysis, based on TGF, accurately reflected the actual sleep duration. However, because of the comparable environmental conditions of our sleep laboratory, this risk was the same in all three groups and should not have influenced the results. Also, patients included in the study received treatment of the main disease. The medications taken by the participants in our study could have interfered with polygraphy results. Glucocorticosteroids, especially in long-term users, are linked to insomnia, increased wakefulness, and sleep architecture alteration by lowering serotonin levels, increasing blood pressure, disrupting glucose metabolism, and developing obesity [38,39,40]. Betahistine increases levels of histamine, acetylcholine, norepinephrine, serotonin, and GABA, but it does not appear to induce sleep apnea and is believed to be safe in patients with OSA [41]. Piracetam reduces sleep attacks in male patients with severe secondary narcolepsy [42], magnesium supplementation improves sleep time and can be recommended for people with insomnia [43], whereas lower vitamin B12 levels have been found to be associated with a higher risk of insomnia [44]. Ondasentron reduced AHI in a dog model but finally the results were not confirmed in humans [3]. The effect of lidocaine infusion on sleep quality was investigated, but no important relationship was reported [45]. In our study medical orders were personalized and varied between each other; some patients received one drug, others several. For this reason, we were unable to describe the exact effect of the substance on sleep patterns, especially since in most cases these drugs were administered in the morning or around noon and only for a few days, which should have minimized the impact on sleep. In addition, we assumed that after disease remission, the brain’s functionality corresponds to the state before the episode of acute idiopathic disease. It was impossible to assess the exact time of the beginning of the neurological process leading to important changes in sleep parameters, and neurological changes may have commenced several nights before the SVA/SSNHL episode.

## 5. Conclusions

Our findings conclude that sudden idiopathic inner ear dysfunction leads to a notable decline in sleep quality, specifically characterized by a significant increase in general and central apnea, hypopnea, and the intensity of snoring. We emphasize the impact of the functional state of the labyrinth on the respiratory system and on maintaining airway patency during sleep. Furthermore, our findings suggest a potential link between chronic vertigo or hearing loss and reduced respiratory-related disruptions during sleep.

## Figures and Tables

**Figure 1 audiolres-15-00076-f001:**
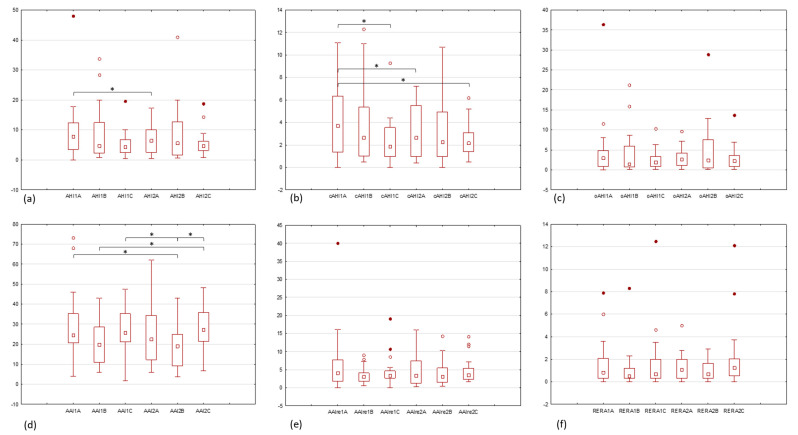
Statistics of recorded sleep parameters: (**a**) AHI, apnea-hypopnea index; (**b**) cAHI, central apnea-hypopnea index; (**c**) oAHI, obstructive apnea-hypopnea index; (**d**) AAI, autonomic arousal index; (**e**) AAIre, autonomic arousal index related to respiratory events; (**f**) RERA, respiratory effort-related arousal index; *1*, first night, *2*, second night measurement; *A*, Group A—acute idiopathic functional inner ear disease, *B*, Group B—chronic functional inner ear disease, *C*, Group C—healthy controls. box: 25–75%, whisker: non-outlier range, □ median, ° outliers, • extreme outliers, * *p* < 0.05.

**Table 1 audiolres-15-00076-t001:** Exclusion criteria.

Criterion	Exclusion
Age	<18 years old
Obesity	BMI ^1^ > 30 kg/m^2^
Sleep disordered breathing	diagnosed or suspected AHI ^2^ ≥ 15/h in any previous sleep study ESS ^3^ score > 15
Comorbidities	mental disorders cardiovascular disorders, e.g., atrial fibrillation, coronary artery disease, arteriosclerosis neurologic disorders, e.g., CVA ^4^, SM ^5^ cranial traumatism (recent or past) pulmonary disorders, e.g., asthma, COPD ^6^
Blood test abnormalities	electrolytes: potassium, natrium complete blood count lipid panel glucose C-reactive protein coagulation screen
Upper airway	nasal septum deviation retrognathia vasomotor or hypertrophic rhinitis enlarged tonsils: 3 or 4 stage Friedman * tongue base or soft palate hypertrophy: III or IV stage Friedman *
Exogenous substances	drugs targeting the central or peripheral nervous system drugs targeting muscles system addiction to psychoactive substances alcohol

In general, patients with a history of previously diagnosed sleep disordered breathing, abnormal blood tests analyses, heart arrhythmia, coronary artery disease, arteriosclerosis, neurological, pulmonary, or psychiatric chronic diseases, and those who systematically used substances targeting the central or peripheral nervous system were disqualified from the study. * Friedman, M.; Salapatas, A.M.; Bonzelaar, L.B. Updated Friedman Staging System for Obstructive Sleep Apnea. Adv. Otorhinolaryngol. 2017, Vol. 80, pp. 41–48 [20]. ^1^ BMI; body mass index, ^2^ AHI; apnea-hypopnea index, ^3^ ESS; Epworth sleepiness score, ^4^ CVA; cerebrovascular accident, ^5^ SM; sclerosis multiplex, ^6^ COPD; chronic obstructive pulmonary disease.

**Table 2 audiolres-15-00076-t002:** Definitions of analyzed parameter.

Parameter	Description
TGF ^1^	Time when the diagnostic nasal cannula provided good signal quality for this length of time.
AHI	Number of apneas and hypopneas per hour within the artefact-free analysis time of the flow signal.
cAHI ^2^	Number of central apneas and hypopneas per hour within the artifact-free evaluation time of the flow signal.
oAHI ^3^	Number of obstructive apneas and hypopneas per hour within the artifact-free evaluation time of the flow signal.
AAI ^4^	Number of autonomous arousals per hour within the artefact-free analysis time of the pulsoximetry signal. This parameter was not displayed in the event of an arrhythmia warning.
AAIre ^5^	Number of autonomous arousals per hour within the artefact-free analysis time of the pulsoximetry signals caused by a respiratory event. This parameter was not displayed in the event of an arrhythmia warning.
RERA ^6^	Number of autonomous arousals per hour within the artefact-free analysis time of the pulsoximetry signals caused by increased respiratory effort. This parameter was not displayed in the event of an arrhythmia warning.
Low risk of sleep fragmentation	AHI: <10 and AAI: <30
Moderate risk of sleep fragmentation	AHI: 10–15 and AAI: 30–40
High risk of sleep fragmentation	AHI: >15 and AAI: >40
Sn ^7^	Proportion of snoring time within the artifact-free evaluation time of the flow signal, measured in %

Definitions of parameters used for evaluation of sleep are according to home sleep apnea testing device (SomnoCheck micro cardio, Weinmann Medical Technology, Hamburg, Germany) settings. ^1^ TGF; time with good flow signal quality, ^2^ cAHI; central apnea-hypopnea index, ^3^ oAHI; obstructive apnea-hypopnea index, ^4^ AAI; autonomic arousal index, ^5^ AAIre; autonomic arousal index related to respiratory events, ^6^ RERA; respiratory effort-related arousal index, ^7^ Sn; snoring.

**Table 3 audiolres-15-00076-t003:** Descriptive statistics of the examined population.

Group Size	Group A	Group B	Group C
number	20	20	20
**Sex**	**Group A**	**Group B**	**Group C**
M:F *	50%:50%	50%:50%	50%:50%
**Age**	**Group A**	**Group B**	**Group C**
mean ± SD	55.1 ± 17.4	52.9 ± 18	43.9 ± 13.4
median	64.5	48	40
range	58	54	43
*p*-value	0.08		
**BMI [kg/m^2^]**	**Group A**	**Group B**	**Group C**
mean ± SD	26.4 ± 3.4	25.4 ± 2.8	26.2 ± 3.3
median	27.1	24.8	27.2
range	10.8	8.37	11.9
*p*-value	0.58		
**ESS**	**Group A**	**Group B**	**Group C**
mean ± SD	6.4 ± 4.1	8.5 ± 4.2	8 ± 4.2
median	6.5	8	8
range	15	13	13
*p*-value	0.27		
**Interval Between Sleep Studies [Number of Days]**	**Group A**	**Group B**	**Group C**
mean ± SD ^1^	4.4 ± 2.9	3.1 ± 1.9	3 ± 2.3
median	3.5	3	2
range	12	5	7
*p*-value	0.13		

Characteristics of the 3 compared groups: Group A—patients with severe idiopathic inner ear dysfunction (sudden vertigo attacks, sudden sensorineural hearing loss), Group B—chronic functional inner ear disease (chronic peripheral vertigo, permanent hearing loss), Group C—healthy population (no otoneurological complaints). The examined groups were comparable in terms of gender, age, BMI, ESS achievements, and time between both sleep studies. * male-to-female ratio. ^1^ SD; standard deviation.

**Table 4 audiolres-15-00076-t004:** Descriptive statistics of sleep time.

TGF [Minutes]	TGF1A	TGF2A	TGF1B	TGF2B	TGF1C	TGF2C
*n* ^1^	20	20	20	20	20	20
mean	388.3	400.7	416.2	418.3	430.8	418.3
SD	108.5	117.7	91.1	95.4	42.6	69.2
difference of the means	13		2		−13	
median	460.5	459	462.5	455.5	449.5	455
range	302	403	324	406	159	243
variance	11768.1	13846.8	8299.3	9104	1817.5	4783.8
*p*-value	0.761 (ns ^2^)		0.609 (ns)		0.629 (ns)	

Parameters were recorded during the night of hospital admission (1) and a few nights later (2) in three groups: A, acute idiopathic functional inner ear disease; B, chronic functional inner ear disease; and C, healthy controls without any otoneurological symptoms. The Wilcoxon signed-rank test was used to compare the two nights in each group. ^1^
*n*; number, ^2^ ns; not significant.

**Table 5 audiolres-15-00076-t005:** Descriptive statistics of sleep apnea-hypopnea indices.

AHI	AHI1A	AHI2A	AHI1B	AHI2B	AHI1C	AHI2C
*n*	20	20	20	20	20	20
mean	9.9	6.8	8.8	9.1	5.2	5.7
SD	10.3	4.4	9.3	9.7	4.3	4.3
difference of the means	−3.1		0.3		0.5	
median	7.9	6.5	4.8	5.6	4.3	4.7
range	48	16.8	33	40.4	19.2	18
variance	106.5	19.2	85.6	95.1	18.6	18.2
*p*-value	0.027		0.955 (ns)		0.409 (ns)	
**cAHI**	**cAHI1A**	**cAHI2A**	**cAHI1B**	**cAHI2B**	**cAHI1C**	**cAHI2C**
*n*	20	20	20	20	20	20
mean	4.1	3	3.9	3.5	2.3	2.4
SD	3.2	2.3	3.7	3.1	2.1	1.4
difference of the means	−1.1		−0.4		0.1	
median	3.7	2.7	2.7	2.7	1.9	2.2
range	11.1	6.8	11.8	10.7	9.3	5.7
variance	10.5	5.2	13.7	9.7	4.4	2
*p*-value	0.011		0.765 (ns)		0.601 (ns)	
**oAHI**	**oAHI1A**	**oAHI2A**	**oAHI1B**	**oAHI2B**	**oAHI1C**	**oAHI2C**
*n*	20	20	20	20	20	20
mean	5	3.1	4.1	5	2.5	2.9
SD	7.9	2.5	5.6	6.8	2.4	3.1
difference of the means	−1.9		0.9		0.3	
median	3	2.6	1.5	2.5	2	2.2
range	36.4	9.6	21.1	28.8	10.2	13.6
variance	63.1	6.4	31.6	46.8	5.8	9.5
*p*-value	0.204 (ns)		0.240 (ns)		0.260 (ns)	

Parameters were recorded during the night of hospital admission (1) and a few nights later (2) in three groups: A, acute idiopathic functional inner ear disease; B, chronic functional inner ear disease; and C, healthy controls without any otoneurological symptoms. The Wilcoxon signed-rank test was used to compare the two nights in each group. The overall AHI and its central components were significantly higher in patients with sudden sensorineural hearing loss or acute vertigo attack when symptoms were present compared to the asymptomatic night after recovery.

**Table 6 audiolres-15-00076-t006:** Descriptive statistics of sleep arousal indices.

AAI	AAI1A	AAI2A	AAI1B	AAI2B	AAI1C	AAI2C
*n*	18	18	19	20	20	20
mean	29.9	25.5	21	18.8	27.3	28.5
SD	17.9	15.4	11.3	9.9	11.3	10.5
difference of the means	−4.4		−2.2		1.2	
median	24.4	22.5	19.8	19.1	25.7	27.3
range	69.3	56.3	37.1	39.4	45.6	41.5
variance	319.3	235.7	126.9	97.5	128.6	110
*p*-value	0.134 (ns)		0.099 (ns)		0.36 (ns)	
**AAIre**	**AAIre1A**	**AAIre2A**	**AAIre1B**	**AAIre2B**	**AAIre1C**	**AAIre2C**
*n*	18	18	19	20	20	20
mean	7	4.8	3.5	4	4.5	4.9
SD	9.4	4.4	2.4	3.4	4.2	3.6
difference of the means	−2.2		0.4		0.4	
median	4.1	3.4	3	3.2	3.4	3.5
range	40	15.7	8.5	13.8	19.1	12.5
variance	88.2	19	5.6	11.6	18	13.3
*p*-value	0.266 (ns)		0.763 (ns)		0.277 (ns)	
**RERA**	**RERA1A**	**RERA2A**	**RERA1B**	**RERA2B**	**RERA1C**	**RERA2C**
*n*	18	18	19	20	19	20
mean	1.7	1.3	1.1	1	1.7	2.1
SD	2.2	1.3	1.9	0.9	2.9	2.9
difference of the means	−0.4		−0.1		0.4	
median	0.9	1.1	0.5	0.7	0.7	1.3
range	7.9	5	8.3	2.9	12.5	12.1
variance	4.8	1.6	3.4	0.8	8.5	8.6
*p*-value	0.955 (ns)		0.234 (ns)		0.22 (ns)	

Parameters were recorded during the night of hospital admission (1) and a few nights later (2) in three groups: A, acute idiopathic functional inner ear disease; B, chronic functional inner ear disease; and C, healthy controls without any otoneurological symptoms. The Wilcoxon signed-rank test was used to compare the two nights in each group.

**Table 7 audiolres-15-00076-t007:** Descriptive statistics of snoring time.

Sn [%TGF]	Sn1A	Sn2A	Sn1B	Sn2B	Sn1C	Sn2C
*n*	15	17	17	18	19	18
mean	5	3.2	4.2	5.4	7.4	6.3
SD	8.4	6.4	9.7	8.1	8.4	7.6
difference of the means	−1.8		1.3		−1.1	
median	1	0	1	1	7	3
range	30	21	40	28	35	27
variance	70.6	40.6	93.5	65	70.1	58.1
*p*-value	0.028		0.410 (ns)		0.394 (ns)	

Parameters were recorded during the night of hospital admission (1) and a few nights later (2) in three groups: A, acute idiopathic functional inner ear disease; B, chronic functional inner ear disease; and C, healthy controls without any otoneurological symptoms. The Wilcoxon signed-rank test was used to compare the two nights in each group.

## Data Availability

The data presented in this study are available on request from the corresponding author due to ethical reasons.

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
