# Peer review of "Sleep Assessment in Patients with Inner Ear Functional Disorders: A Prospective Cohort Study Investigating Sleep Quality Through Polygraphy Recordings"

_audiolres, 2025, doi:10.3390/audiolres15040076_

Round 1

Reviewer 1 Report

Comments and Suggestions for Authors

REVEWI OF SLEEP ASSESMENT IN PATIETS WITH INNNER EAR FUCNTIONAL DISORDERS. A PROSPECTIVE COHOT STUDY INVESTIGATING SLEEP QUALITY THROUGH POLYGRAPHY RECORDING.

 The authors hypothesized that acute inner ear disorders would cause dysautonomia with impaired breathing adjustment, changes venous return and could lead to orthostatic hypotension. To study this hypothesis, they prospectively recorded polysomnograms in 60 participants (examined in the acute phase and subsequently within 14 months) There were three groups: Sudden hearing loss or sudden vertigo 2. Chronic ear disease and 3. Normal healthy population.

Interestingly, they found a significant increase in apnea and hypopnea (central type) in those patients with an “acute event”. These parallels the observations in the absence of gravity in space, astronauts need to take sleeping pills. In healthy individuals’ stimulation of the vertical SCC increases respiratory rate.

This is an interesting study and raises a question about the significance of central lesions causing vertigo, particularly when located in the brainstem.

An important point that the authors must make is the effect of medications aa a potential confounder. Acute hearing loss patients usually take steroids and acute vertigo patients receive antihistamine medication. Bithi of these medications will affect vigilance and change sleep .

Reviewer 2 Report

Comments and Suggestions for Authors

I have read the article audiolres-3656645 entitled:

Sleep Assessment in Patients with Inner Ear Functional Disorders: A Prospective Cohort Study Investigating Sleep Quality through Polygraphy Recordings ».

This prospective article intends  to exlore the vestibulo-respiratory reflex  by analyzing 60 patients divided in 3 cohorts of 20 patients

This work  was conducted in accordance with the Declaration of Helsinki with an Informed Consent Statement and provided with  an Institutional Review Board Statement: approved by the Ethics Committee of the Centre of Postgraduate Medical.

The reported polygraphic explorations were performed  in 60 patients divided in 3 groups ( group A=20 cases with sudden idiopathic vertigo attack and/or sudden HL; group B= 20 cases chronic peripheral vertigo and/or Chronic HL; group C= 20 healthy individuals). Polygraphic recording were performed twice: in the group A at the initial attack and then at J1-13 later. In group B initially and then 1-6 days later and for group C the delay between 2 recording was 1-8 days.

The studied parameters were 1) Apnea hypopnea index  (AHI). 2)Central apnea hypopnea index (cAHI). 3)Obstructive apnea-hypopnea Index (oAHI) . 4) Autonomic Arousal ( AAI). 5) Autonomic arousal Index related to respiratory events (AAIre). 6)Respiratory effort related arousal Index ( RERA)

The results showed a significant increase in AHI and even more in cAHI in the patients with acute inner ear dysfunction

Patients of group A experienced more arousal during the acute inner ear dysfunction. Results are significantly different from Chronic vestibular disorders (group B) and from the control group C.

This article is well written; however the patient series is small (20 for each subgroup)

In the section participants in the table 1:  (Exclusion criteria) The exclusion of a context of cranial traumatism ( CT) should be also mentioned. CT may influence Orexine A &B secretion (Hypocretine) and modify  polygraphy. The acute idiopathic vestibular disorders included in this series fortunately exclude labyrinthine commotion or traumatisms.

Otherwise the modifications observed is possibly not specific of a vestibular disease  and one may regret that these results were not compared to those in patients with a different acute stress such as , a cranial traumatism, or  small body injury or anxious stress which is possibly able to interfere in a not much different but not specific manner with sleep discomfort. In another words can any other health acute disorder modify similarly polysomnography without including specific modifications? This absence of specificity could be mentioned in the discussion.  Modifications of orexin ( hypocretine) are also observed in pure central disease (neurological or degenerative..) or after cranial traumatism.

The discussion could be shortened and clarified

The conclusion in the abstract is possibly questionable in its formulation: “…deterioration in sleep quality in contrast to individuals with chronic inner ear impairments, who exhibit better sleep quality than  healthy individuals, potentially due to central plasticity and compensatory-adaptive mechanisms” It is probably not the central plasticity which provides better sleep quality than healthy individuals. This sentence should be reformulated to rule out any ambiguity or misunderstanding.. Otherwise this assertion in chronic vestibular diseases is not corroborated in other works ( Martin et al 2016)(chronic idiopathic  bilat vestibular loss).

Small detail in the references . The date (year) of publication of each article is written in bold figures except ref 30 (line 454). This could be homogenized.

This interesting article raises the question of the relationship between vestibular disease and quality of sleep by describing the  vestibulo-respiratory reflex. Some assertions could be clarified or rephrased in the abstract conclusion and the discussion improved.
